# Network Analysis-Based Disentanglement of the Symptom Heterogeneity in Asian Patients with Schizophrenia: Findings from the Research on Asian Psychotropic Prescription Patterns for Antipsychotics

**DOI:** 10.3390/jpm12010033

**Published:** 2022-01-03

**Authors:** Joonho Choi, Hyung-Jun Yoon, Jae Hong Park, Yukako Nakagami, Chika Kubota, Toshiya Inada, Takahiro A. Kato, Shu-Yu Yang, Sih-Ku Lin, Mian-Yoon Chong, Ajit Avasthi, Sandeep Grover, Roy Abraham Kallivayalil, Andi Jaylangkara Tanra, Kok Yoon Chee, Yu-Tao Xiang, Kang Sim, Afzal Javed, Chay Hoon Tan, Norman Sartorius, Shigenobu Kanba, Naotaka Shinfuku, Yong Chon Park, Seon-Cheol Park

**Affiliations:** 1Department of Psychiatry, Hanyang University College of Medicine, Seoul 04763, Korea; jchoi@hanyang.ac.kr (J.C.); hypyc@hanyang.ac.kr (Y.C.P.); 2Department of Psychiatry, Hanyang University Guri Hospital, Guri 11923, Korea; 3Department of Psychiatry, College of Medicine, Chosun University, Gwangju 61452, Korea; YoonHyungJun@chosun.ac.kr; 4Department of Psychiatry, College of Medicine, Dong-A University, Busan 47392, Korea; prozac620@hanmail.net; 5Department of Psychiatry, Kyoto University Graduate School of Medicine, Kyoto 606-8501, Japan; nakagami.yukako.4s@kyoto-u.ac.jp; 6Department of Psychiatry, National Center of Neurology and Psychiatry, Tokyo 187-8551, Japan; kubotachika@ncnp.go.jp; 7Department of Psychiatry, Nagoya University Graduate School of Medicine, Nagoya 466-8550, Japan; toshiya.inada@gmail.com; 8Department of Neuropsychiatry, Graduate School of Medical Sciences, Kyushu University, Fukuoka 819-0395, Japan; takahiro@npsych.med.kyushu-u.ac.jp (T.A.K.); skanba@npsych.med.kysuhu-u.ac.jp (S.K.); 9Department of Pharmacy, Taipei City Hospital, Fu Jen University, Taipei 24205, Taiwan; shuyu@ksts.seed.net.tw; 10School of Medicine, College of Medicine, Fu Jen Catholic University, New Taipei 242062, Taiwan; 11Department of Psychiatry, Linkou Chang Gung Memorial Hospital, Taoyuan 33305, Taiwan; DAF68@tpech.gov.tw; 12Department of Psychiatry, Kaohsiung Chang Gung Memorial Hospital, Kaohsiung & Chang Gung University School of Medicine, Taoyuan 83301, Taiwan; chongmy@live.com; 13Department of Psychiatry, Post Graduate Institute of Medical Education and Research, Chandigarh 160012, India; drajitavasthi@yahoo.co.in (A.A.); drsandeepg2002@yahoo.com (S.G.); 14Department of Psychiatry, Pushpagiri Institute of Medical Sciences, Tiruvalla 689101, India; roykalli@gmail.com; 15Department of Psychiatry, Faculty of Medicine, Hasanuddin University, Makassar 90245, Indonesia; ajtanra@yahoo.com; 16Tunku Abdul Rahman Institute of Neuroscience, Kuala Lumpur Hospital, Kuala Lumpur 50586, Malaysia; cheekokyoon@yahoo.com; 17Unit of Psychiatry, Department of Public Health and Medicinal Administration, Institute of Translational Medicine, Faculty of Health Sciences, University of Macau, Macao SAR 820006, China; xyutly@gmail.com; 18Institute of Mental Health, Buangkok Green Medical Park, Singapore 539747, Singapore; ksim6133@gmail.com; 19Pakistan Psychiatric Research Centre, Fontain House, Lahore 1317, Pakistan; afzalj@gmail.com; 20Department of Pharmacology, National University of Singapore, Singapore 119244, Singapore; phctanch@nus.edu.sg; 21Association for the Improvement of Mental Health Programmes, 1211 Geneva, Switzerland; sartorius@normansartorius.com; 22Department of Social Welfare, School of Human Sciences, Seinan Gakuin University, Fukuoka 814-8511, Japan; shinfukunaotaka@gmail.com

**Keywords:** Brief Psychiatric Rating Scale, heterogeneity, motor retardation, network analysis, precision medicine, schizophrenia

## Abstract

The symptom heterogeneity of schizophrenia is consistent with Wittgenstein’s analogy of a language game. From the perspective of precision medicine, this study aimed to estimate the symptom presentation and identify the psychonectome in Asian patients, using data obtained from the Research on Asian Psychotropic Prescription Patterns for Antipsychotics. We constructed a network structure of the Brief Psychiatric Rating Scale (BPRS) items in 1438 Asian patients with schizophrenia. Furthermore, all the BPRS items were considered to be an ordered categorical variable ranging in value from 1–7. Motor retardation was situated most centrally within the BPRS network structure, followed by depressive mood and unusual thought content. Contrastingly, hallucinatory behavior was situated least centrally within the network structure. Using a community detection algorithm, the BPRS items were organized into positive, negative, and general symptom clusters. Overall, DSM symptoms were not more central than non-DSM symptoms within the symptom network of Asian patients with schizophrenia. Thus, motor retardation, which results from the unmet needs associated with current antipsychotic medications for schizophrenia, may be a tailored treatment target for Asian patients with schizophrenia. Based on these findings, targeting non-dopamine systems (glutamate, γ-aminobutyric acid) may represent an effective strategy with respect to precision medicine for psychosis.

## 1. Introduction

In psychiatric taxonomy, schizophrenia has been conceptualized as a unitary disease entity. However, the etiopathology, symptomatology, and clinical courses of schizophrenia are heterogeneous [1,2,3]. To explain the heterogeneity of the clinical features of schizophrenia, several models describing the etiology, pathophysiology, and illness have been proposed. Regarding the traditional ‘single common pathway’ construct, it is presumed that the interactions among multiple etiological factors produce neurobiological alterations, which lead to the broadly similar phenotypic expression of ‘single’ schizophrenia through the ‘final common pathophysiological process.’ Contrarily, in the ‘single schizophrenia with many domains’ model, it is presumed that the interactions among multiple etiological factors lead to the multiple concurrent pathophysiological dimension. Moreover, in the ‘many schizophrenias’ model, it is presumed that the interactions among multiple etiological factors lead to different types of the disease with a similar array of symptoms. Thus, precision medicine is required to disentangle the symptom heterogeneity of schizophrenia [2].

Regarding the symptom heterogeneity of schizophrenia, its definitions and boundaries have been changed based on the influences of available diagnostic methods and therapeutic modalities in the past century [3]. The concept of schizophrenia has evolved from Emil Kraepelin’s nosological principle to the description of schizophrenia in the fifth edition of the Diagnostic and Statistical Manual of Mental Disorders (DSM-5) as follows: Kraepelin [4] incorporated hebephrenia, catatonia, and paranoia into a single disease entity of *dementia praecox* (schizophrenia) since he had noticed that all the patients with hebephrenia, catatonia, and paranoia shared the similar clinical course and outcome, including adolescent or early adult onset, tendency towards deterioration, and an outcome of dementia. Moreover, he made a distinction of *dementia praecox* from *folie circulaire* (manic-depressive insanity), which had characteristic features, including episodicity, absence of deterioration, and a more favorable outcome [4,5]. Under the Kraepelinian nosological principle, *Zerfahrenheit*, which denoted the loss of internal or external connections of the chains of ideas or loss of rational ties between the associations, was proposed as a typical form of thought and language disorder in *dementia praecox* [5]. Whereas Kraepelin emphasized the importance of the longitudinal course and outcome in defining schizophrenia, both Eugen Bleuler and Kurt Schneider proposed specific symptom criteria [3]. Bleuler [6] stated that not delusion and hallucination but a set of symptoms including loosening of associations, blunted affect, ambivalence, and autism comprise the basic or fundamental symptoms of schizophrenia. Additionally, Schneider [7] defined the 11 first-rank symptoms as the basis of the definition of ‘nuclear’ schizophrenia. Since Bleulerian viewpoints broadly prevailed in the USA by the 1960s, ‘loss of ego boundaries’ was defined as the basic symptom of the DSM-II criteria for schizophrenia. Thus, DSM-II [8] provided the broadest definition of schizophrenia in a historical framework. Contrastingly, the narrowest definition of schizophrenia was included in DSM-III [9]. The boundaries of schizophrenia have been modestly expanded in subsequent editions of the DSM (DSM-III-R, DSM-IV, and DSM-IV-TR) as a reaction to its narrowest definition. It is presumed that the DSM concept of schizophrenia has been defined by an integration of Kraepelinian chronicity, Bleulerian negative symptoms, and Schneiderian positive symptoms [3]. However, this concept has been criticized in terms of the regression to pre-Kraepelinian nosology because its definition has narrowed down into a predominantly chronic delusional-hallucinatory syndrome with the disappearance of non-paranoid schizophrenia (hebephrenia) and oblivion of the constitutive ideas with respect to its psychopathological nature [10]. As a reaction to the Kraepelinian nosological principle, deconstructing schizophrenia or defining psychosis with dimensions and intermediated phenotypes have been proposed [11]. Thus, the DSM-5 concept for schizophrenia has been defined predominantly based on the categorical concept, with the help of the dimensional concept [12].

The current concept of schizophrenia is considered the discrete kind or fuzzy one partly inconsistent of the Kraepelinian nosological principle because its category boundary and essence are still arbitrary and unclear, respectively [13]. The current concept of schizophrenia is constituted not only of DSM symptoms (i.e., delusion, hallucinations, psychomotor disturbance, bizarre thoughts, and negative symptoms) but also non-DSM symptoms (i.e., impaired reality testing, impaired cognition, social withdrawal, depression, and anxiety). Herein, it is presumed that the symptom heterogeneity of schizophrenia is consistent not with the ‘disease essentialism paradigm’ but with the Wittgensteinian analogy of language game as described below [14]:

‘Consider for example the proceedings that we call games. I mean board-games, card-games, ball-games, Olympic games, and so on. What is common to them all?—Don’t say: ‘There must be something common, or they would not be called games’—but look and see whether there is anything common to all.—For if you look at them you will not see something that is common to all, but similarities, relationships, and a whole series of them at that. To repeat: don’t think, but look! … the concept game is a concept with blurred edges.—‘But is a blurred concept a concept at all?’—Is an indistinct photograph a picture of a person at all? Is it even always an advantage to replace an indistinct picture by a sharp one? Isn’t the indistinct one often exactly what we need? [15]’.

In other words, since the cases of schizophrenia are connected by ‘family resemblance’, but not by ‘essence’, the category of schizophrenia is regarded as an operational constitute but not a natural kind [14,16]. Thus, Thomas Insel’s ‘next-generation treatment for mental disorders’ [17,18] can be considered as an alternative approach to disentangle the symptom heterogeneity of schizophrenia. Consistent with the change in the theoretical construct from a chemical imbalance to dysfunctional circuitry, it has been suggested that clinical targets should be changed from the clinical diagnoses (e.g., psychosis, mood disorder, and anxiety disorder) to symptoms or endophenotypes (e.g., amotivation, attentional bias, executive function, anhedonia, social deficit, and working memory). Precision medicine includes tailoring treatments to a specific disease process and parsing the etiology or underlying disease mechanism-based heterogeneous syndrome. Understanding the distinct etiology of a disease process is required to split complex syndromes into etiologically homogeneous subtypes in terms of precision medicine targeting not broad-based but specific etiologies [19]. Regarding precision medicine, differences in the clinical features of the research sample may contribute to hampering the identification of clinically valid and reliable biomarkers of psychosis [20]. A ‘psychonectome’ has been proposed as a complex ensemble of dependencies between psychological constructs to formalize the idea of psychological constructs forming a dynamic network of mutually dependent elements [21]. Herein, it has been proposed that identifying a psychonectome for the symptom heterogeneity of schizophrenia can help establish precision medicine as an efficient intervention for psychosis. Notably, a network analysis may comprise a novel computational method to disentangle the symptom heterogeneity of schizophrenia [22]. Based on the determination of the variables contributing proportionally or disproportionally to the adaptive functioning of the network, the symptom heterogeneity of schizophrenia can be adequately evaluated within an estimated network structure [23]. Since centrality is defined as the overall connectivity of an individual symptom within a network structure, it is suggested that the central symptoms can contribute to the rapid activation of the interrelated symptoms within the network structure and comprise the potential therapeutic targets [24,25]. A network analysis is based on the idea that “symptoms are not outcome factors of an underlying disease; symptoms and the associations between them are the disease itself” [26]. This study aimed to estimate a network structure and identify a psychonectome from the symptom heterogeneity of schizophrenia, using data obtained from the Research on Asian Psychotropic Prescription Patterns for Antipsychotics (REAP-AP) [27,28].

## 2. Materials and Methods

### 2.1. Study Overview and Participants

As described elsewhere, the aims of the REAP-AP [27,28], which was one of the largest international research collaborations in Asian countries or special administrative areas, were to examine patterns of psychotropic drug use and their related clinical characteristics, as well as to explore ways of improving psychotropic drug use in Asian patients with schizophrenia. In total, 3744 consecutive patients with schizophrenia were enrolled by the 4th REAP-AP between March and June 2016, from 71 survey centers in 15 Asian countries and areas (Bangladesh, China, Hong Kong, India, Indonesia, Japan, South Korea, Malaysia, Myanmar, Pakistan, Singapore, Sri Lanka, Taiwan, Thailand, and Vietnam). The study protocol and informed consent forms were approved by the institutional review boards of Taipei City Hospital, Taipei, Taiwan (receipt number: TCHIRB-10412128-E) and other hospitals participating in the survey. All the study participants signed informed consent forms prior to participation. Since short or long case report forms could be used depending on the resources available to the participating countries or special administrative areas, the 18-item Brief Psychiatric Rating Scale (BPRS) [29] was used. Thus, in this study, we used only data from the participants who met the following inclusion criteria: (i) diagnosis of schizophrenia, based on DSM-5 [12], by clinical psychiatrists at survey centers, (ii) medication with antipsychotics, coded as the F05A under the Anatomical Therapeutic Chemical (ATC) classification system [30], (iii) age ≥18 and ≤80 years, and (iv) availability of the complete 18-item BPRS [29]. Furthermore, we excluded data from participants who met the following exclusion criteria: (i) comorbidity of organic mental disorders, bipolar disorders, or intellectual disorder; and (ii) comorbid seizure disorders, other neurological diseases, and severe physical disease. Finally, 1438 patients with schizophrenia were enrolled from five Asian countries, including India, Indonesia, Japan, Malaysia, and Taiwan.

### 2.2. Brief Psychiatric Rating Scale

We used the 18-item BPRS to evaluate the psychiatric symptoms of the participants. All the BPRS items were scored on a 7-point Likert scale from ‘not present’ (1) to ‘very severe’ (7). Its psychometric properties included reliability and validity in addition to others [29,31,32]. Its English version was commonly used by clinical psychiatrists and study coordinators at the survey centers because the study participants were enrolled from five Asian countries with different languages. Based on the DSM-5 criteria for schizophrenia, the BPRS items were divided into DSM symptoms (i.e., emotional withdrawal (EMO), conceptual disorganization (CON), mannerism and posturing (MAN), suspiciousness (SUS), hallucinatory behavior (HAL), motor retardation (MOT), unusual thought content (THO), and blunted affect (BLU)) and non-DSM symptoms (i.e., somatic concern (SOM), anxiety (ANX), guilty feelings (GUI), tension (TEN), grandiosity (GRA), depressive mood (DEP), hostility (HOS), uncooperativeness (UNC), excitement (EXC), and disorientation (DIS)).

### 2.3. Operational Classification of Psychotropic Drugs

Using the ATC classification system [30], psychotropic drugs were classified into antipsychotics (N05A), mood stabilizers (antiepileptics and lithium; N03A and N05AN), antidepressants (N06A), anxiolytics (N05B), hypnotics (N05C), and antiparkinsonian drugs (N04). Although lithium and clonazepam were defined as antipsychotics and antiepileptics, respectively, under the ATC classification system, they were classified as a mood stabilizer and a hypnotic, respectively, on the basis of conventional grouping. High-dose antipsychotics were operationally defined as either a chlorpromazine equivalent corresponding to a cumulative dose of ≥1000 mg/day [33] or a ratio of prescribed daily dose to defined daily dose ≥1.5 [34].

### 2.4. Statistical Analyses

Using the R-package qgraph [35], a network structure, which consisted of both nodes (corresponding to symptoms) and edges (corresponding to associations among symptoms), was estimated for 18 items of the BPRS. All the BPRS items were considered to be ordered-categorical variables ranging in a value from 1–7. Due to the cross-sectional characteristic of our data, the network structure was estimated in a unidirectional manner. Network analyses were based on polychoric correlations. Using the least absolute shrinkage and selection operator (LASSO) [36], false-positive edges were controlled, and very small edges were set exactly to zero. Using the graphical LASSO (GLASSO) procedure, since the edges were defined as partial correlation coefficients, the average edge was defined based on the relationship level between two symptoms controlling for all other relationships within the network. Using the shrinkage parameter, the extended Bayesian Information Criterion [37] was minimized, and the underlying network structures were accurately recovered [25]. We used the Frutchterman-Reingold algorithm [38] to place stronger connected nodes closer together with an estimated network structure. Moreover, using a modularity-based community-detecting algorithm, we investigated whether nodes were clustered together within the estimated network structure. We used the spin-glass community algorithm [39] to test whether the number and weighted strength of edges within a cluster exceeded those within another cluster in terms of communities within the network (weights = null, vertex = null, parupdate = false, gamma = 0.5, start temperature = 1, stop temperature = 0.01, cooling factor = 0.99, spins = 17).

Regarding the node statistics, the centrality of all the BPRS items was estimated as follows [40]: Node strength centrality, which was a common and stable metric, denoted the sum of all associations of a given node with all the other nodes. Betweenness centrality denoted the shortest length of a path connecting any two nodes. Closeness centrality denoted the measure of how close a symptom was to all other symptoms. Since node strength centrality was substantially correlated with betweenness centrality and closeness centrality, the most central symptoms within the network structure of all the BPRS items were estimated based on the node strength centrality, betweenness centrality, and closeness centrality. Using a permutation test [41], the node statistics across different symptom groups (i.e., DSM and non-DSM symptoms) were compared. By assigning symptoms randomly to the two groups 100,000 times, the difference between the groups at each time was estimated. If the difference between the two groups was observed within 2.5% on either side of the distribution, the test significance was set at *p* < 0.05. Using a correlation stability coefficient (CS-coefficient), the centrality stability was operationally defined since the CS-coefficient denotes the maximum proportion of cases that can be eliminated to obtain a 95% probability that the ranking correlation between the original network and case-subset network would amount to a very large effect (0.7) [42]. Thus, solely interpreting centrality indices with a CS-coefficient >0.25, but preferentially >0.5, was recommended [43]. Using 95% nonparametric bootstrap confidence intervals (1000 bootstrap samples) of differences between each pair of centrality indices, significant differences between centrality indices were identified.

## 3. Results

### 3.1. General Description of the Study Participants

As shown in Table 1, the cohort consisted of Indian (*n* = 400, 27.8%), Indonesian (*n* = 261, 18.2%), Japanese (*n* = 98, 6.8%), Malaysian (*n* = 299, 20.8%), and Taiwanese (*n* = 380, 26.4%) participants. Approximately half of them were male (*n* = 830, 57.7%) and had a duration of illness >10 years (*n* = 788, 54.8%), and 46.2% of the cohort were inpatients (*n* = 664). Additionally, approximately one-third of them had a duration of untreated psychosis >1 year (*n* = 422, 29.4%). The mean age was 39.9 (standard deviation [SD] = 12.5) years. Regarding the patterns of psychotropic drug use, approximately one-third were treated with antipsychotic polypharmacy (*n* = 536, 37.3%) and adjunctive antiparkinsonian drugs (*n* = 580, 38.9%). Table 2 lists the response frequency distributions of BPRS items.

### 3.2. Edge Statistics

As shown in Figure 1, the construction of a network of the 18 BPRS items revealed that 102 (66.7%) out of 153 possible edges were estimated to be >0. Several interconnections including SOM–ANX (weight = 0.463), EMO–BLU (weight = 0.406), HOS–UNC (weight = 0.397), ANX–TEN (weight = 0.352), GUI–DEP (weight = 0.342), HOS–SUS (weight = 0.329), and others were revealed within the network. The 18 BPRS items were organized into three meaningful clusters by the community-detection analysis. Cluster A consisted of CON–MAN–GRA–HOS–SUS–HAL–UNC–THO–EXC–DIS, cluster B consisted of EMO–MOT–BLU, and cluster C consisted of SOM–ANX–GUI–TEN–DEP.

### 3.3. Node Statistics

As shown in Figure 1 and Figure 2, regarding the inspection of the node strength centrality of BPRS items, MOT was the most centrally situated BPRS item within the network, followed by DEP, THO, EXC, ANX, and HOS. Contrastingly, HAL was the most poorly interconnected BPRS item within the network, followed by SOM, TEN, DIS, EMO, and UNC. Node strength centrality revealed an interpretable level of CS-coefficient (0.361), whereas betweenness centrality and closeness centrality revealed low levels of CS-coefficients (0.128 and 0.206). As shown in Figure 3, there were no fundamental differences between DSM and non-DSM symptoms regarding the difference tests analyzing node strength centrality (*p* = 0.814), betweenness centrality (*p* = 0.831), and closeness centrality (*p* = 0.758).

## 4. Discussion

In summary, motor retardation (MOT), depressive mood (DEP), and unusual thought content (THO) were estimated as the top three central symptoms within the network structure, followed by excitement (EXC), anxiety (ANX), and hostility (HOS). Contrarily, hallucinatory behavior (HAL) was the most poorly interconnected BPRS item within the network structure. Furthermore, overall, DSM symptoms were not more central than non-DSM symptoms within the BPRS items network of Asian patients with schizophrenia. Moreover, 18 BPRS items were organized into three meaningful symptom clusters, including the positive (CON–MAN–GRA–HOS–SUS–HAL–UNC–THO–EXC–DIS), negative (EMO–MOT–BLU), and general symptom clusters (SOM–ANX–GUI–TEN–DEP). Finally, SOM–ANX, EMO–BLU, CON–THO, HOS–UNC, and ANX–TEN interconnections were the top five strongest associations within the networks, followed by MOT–BLU, HOS–SUS, and MAN–EXC interconnections.

It has been presumed that central symptoms may comprise the pharmacological therapeutic targets because central symptoms can contribute to the rapid activation of intertwined symptoms within the network [24,25]. It has been proposed that psychomotor retardation and its biochemical modulation are regarded as the paradigmatic example of a dimensional approach in the Research Domain Criteria. It has been known that psychomotor retardation is neurobiologically underpinned by three mechanisms, including (i) the modulation of substantia nigra-based subcortical–cortical motor circuit primarily by the non-motor subcortical raphe nucleus via the basal ganglia, (ii) modulation of the motor network by non-motor cortical networks such as default-mode and sensory networks, and (iii) shaping the regional distribution of neural activity within the motor cortex by global cortical activity. Moreover, it has been suggested that the operation of psychomotor mechanisms can be performed in a dimensional and transdiagnostic manner, not based on the diagnostic category but on the levels of psychomotor activity [44]. According to findings from previous neuroimaging studies, it has been proposed that abnormalities in the sensorimotor domain are related to the dysfunction of the cerebello-thalamo-cortico-cerebellar network [45]. Additionally, it has been reported that psychomotor slowing is positively related to negative symptoms and mania, regardless of the diagnostic category, among individuals with schizophrenia, schizoaffective disorder, bipolar disorder, and others [46]. Thus, it can be proposed that motor retardation may be the dimensionally defined core constitute of the psychonectome underpinned by the neural network (i.e., the cerebello-thalamo-cortico-cerebellar network) in Asian patients with schizophrenia. It has been speculated that motor retardation may be consistent with the unmet needs of antipsychotic medications for schizophrenia [47]. Thus, from the perspective of precision medicine, targeting the non-dopaminergic systems (glutamate and γ-aminobutyric acid) should be considered [48]. In addition to these networks, the pulvinar nucleus of the thalamus is an important neural structure that can be involved in cognitive, sensory, and motor deficits in patients with schizophrenia. The pulvinar nucleus is the largest nucleus in the thalamus and is mutually connected to several cortical and subcortical regions, including the prefrontal cortex, sensory cortex, superior colliculus, and amygdala [49]. Thus, the pulvinar nucleus plays an important role in normal multisensory processing, emotional response, and decision making, which are significantly impaired in patients with schizophrenia [50,51,52]. A positive relationship has been reported between performance on working memory and activation in the pulvinar nucleus and other structures [53]. Herein, structural and functional abnormalities of the pulvinar nucleus may be a neurobiological underpinning for the motor retardation-centered symptom networks observed in our study. Moreover, it has been reported that the measure of depressive mood is related to the neurobiological dysfunction for reward prediction in transnosological samples, including patients with schizophrenia, alcohol dependence, major depression, bipolar disorder, and attention-deficit/hyperactivity disorder [54]. Further, unusual thought content, in addition to conceptual disorganization and difficulty in abstract thinking, is regarded as a symptom combination predictor for treatment-resistant schizophrenia in clinical practice [55]. Hallucinations comprise one of the hallmark symptoms of schizophrenia and an important treatment target. [56,57] It has also been reported that antipsychotics are a rapid and efficient intervention for hallucinations. Notably, the differential antihallucinatory effects of olanzapine, amisulpride, and aripiprazole have been demonstrated previously [58]. Since the duration of illness in approximately half of the study participants was >10 years in our study, it can be speculated that the hallucinatory behaviors may be the most interconnected BPRS item within the estimated network structure. Since DSM symptoms were not more central than non-DSM symptoms within the BPRS items network, it can be speculated that not only DSM symptoms but also non-DSM symptoms may be considered as therapeutic targets. Moreover, the organization of BPRS items into positive, negative, and general symptom clusters may help match the specific treatment to the relevant symptom cluster.

Our study has several limitations. First, the inter-rater reliability to assess clinical characteristics including the 18-item BPRS was not measured. Second, since our data were collected in a cross-sectional manner, the networks were unidirectionally estimated. However, the differentiation between out-degree centrality and in-degree centrality can be allowed for longitudinal studies. Third, the duration of illness of approximately half of the participants was >10 years. It cannot be excluded that the chronic features of schizophrenia can potentially influence the network structure of the BPRS items. Fourth, the floor and ceiling effects on the network structure cannot be excluded. However, the Pearson correlation coefficient (r) between the standard deviation and node strength centrality of the BPRS items was negligible (−0.12). A potential modified 18-item BPRS, which has been proposed by Sawamura and colleagues [59], can be used in further network analysis studies to overcome these floor and ceiling effects.

## 5. Conclusions

Despite the limitations, our findings can help estimate the network structure of the BPRS items to disentangle the heterogeneity of symptom presentation in Asian patients with schizophrenia. Notably, our findings indicate that motor retardation, which is underpinned by non-dopaminergic systems (glutamate, γ-aminobutyric acid), may be an important therapeutic target for individuals with schizophrenia.

## Figures and Tables

**Figure 1 jpm-12-00033-f001:**
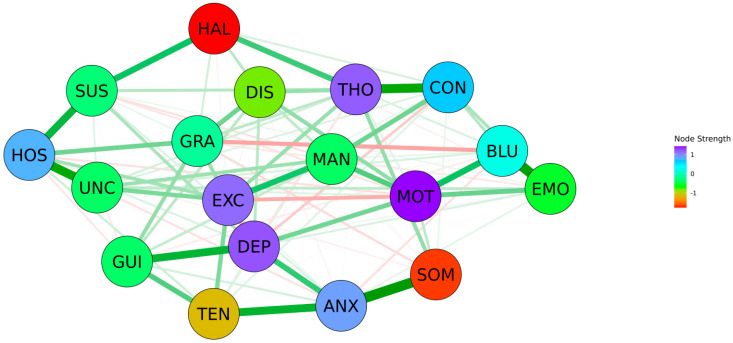
Network structure of the Brief Psychiatric Rating Scale items in Asian patients with schizophrenia (*n* = 1438; CS-coefficient = 0.361). Green lines represent positive associations, whereas red lines represent negative associations between the connecting nodes. The thickness of the lines represents the strength of the edges. Abbreviations: ANX, anxiety; BLU, blunted affect; CON, conceptual disorganization; DEP, depressive mood; DIS, disorientation; EMO, emotional withdrawal; EXC, excitement; GRA, grandiosity; GUI, guilt feelings; HAL, hallucinatory behavior; HOS, hostility; MAN, mannerism and posturing; MOT, motor retardation; SOM, somatic concern; SUS, suspiciousness; THO, unusual thought content; UNC, uncooperativeness.

**Figure 2 jpm-12-00033-f002:**
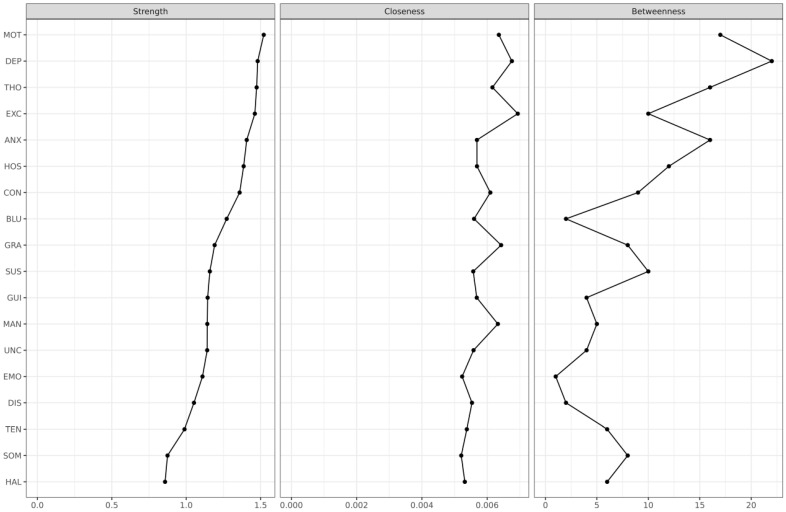
Node statistics including node strength centrality, closeness centrality, and betweenness centrality (*n* = 1438). Abbreviations: ANX, anxiety; BLU, blunted affect; CON, conceptual disorganization; DEP, depressive mood; DIS, disorientation; EMO, emotional withdrawal; EXC, excitement; GRA, grandiosity; GUI, guilt feelings; HAL, hallucinatory behavior; HOS, hostility; MAN, mannerism and posturing; MOT, motor retardation; SOM, somatic concern; SUS, suspiciousness; THO, unusual thought content; UNC, uncooperativeness.

**Figure 3 jpm-12-00033-f003:**
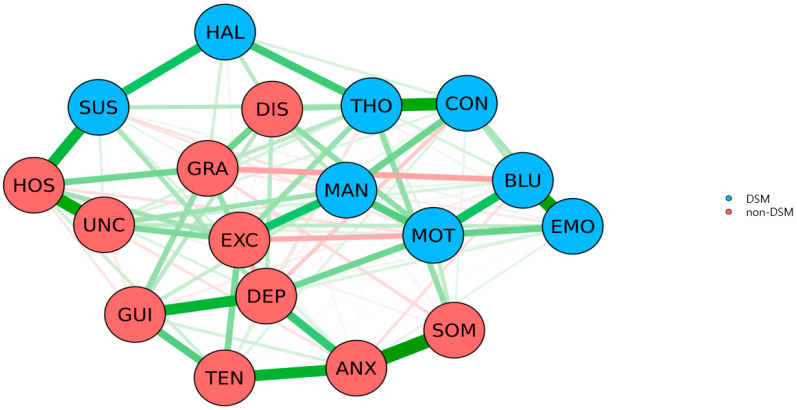
Comparison of node statistics of DSM symptoms and non-DSM symptoms within the network structure of BPRS items in Asian patients with schizophrenia (*n* = 1438; Difference test *p*-value = 0.814). Green lines represent positive associations, whereas red lines represent negative associations between the connecting nodes. The thickness of the lines represents the strength of the edges. Abbreviations: ANX, anxiety; BLU, blunted affect; CON, conceptual disorganization; DEP, depressive mood; DIS, disorientation; EMO, emotional withdrawal; EXC, excitement; GRA, grandiosity; GUI, guilt feelings; HAL, hallucinatory behavior; HOS, hostility; MAN, mannerism and posturing; MOT, motor retardation; SOM, somatic concern; SUS, suspiciousness; THO, unusual thought content; UNC, uncooperativeness.

**Table 1 jpm-12-00033-t001:** General description of the study participants (*n* = 1438).

Continuous Variable	Mean	SD
Age (years)	39.9	12.5
Chlorpromazine equivalent dose (mg/day)	501.5	396.5
Discrete variable	*n*	%
Sex		
Male	830	57.7
Female	608	42.3
Country		
India	400	27.8
Indonesia	261	18.2
Japan	98	6.8
Malaysia	299	20.8
Taiwan	380	26.4
Duration of illness		
<3 months	52	3.6
3–6 months	38	2.6
6–12 months	46	3.2
1–5 years	274	19.1
5–10 years	240	16.7
10–20 years	429	29.8
>20 years	359	25.0
Duration of untreated psychosis		
<3 months	524	36.4
3–12 months	492	34.2
1–5 years	270	18.8
>5 years	152	10.6
Inpatient	664	46.2
Unemployed	237	16.5
Antipsychotic polypharmacy	536	37.3
Adjunctive mood stabilizer	142	9.9
Adjunctive antidepressant	129	9.0
Adjunctive antiparkinsonian	560	38.9
High dose antipsychotic ^†^	161	11.2
Long-acting injectable antipsychotic	305	21.2
Clozapine	293	20.4
Electroconvulsive therapy	42	2.9
Cannabis use (lifetime)	119	8.3

^†^ Cumulative dose of ≥1000 mg/day chlorpromazine equivalent or a ratio of prescribed daily dose (PDD) to the defined daily dose (DDD) ≥1.5. SD, standard deviation.

**Table 2 jpm-12-00033-t002:** Mean (SD) and % score of the Brief Psychiatric Rating Scale items (*n* = 1438).

Items	Abbreviation	Mean (SD)	1	2	3	4	5	6	7
			*n* (%)	*n* (%)	*n* (%)	*n* (%)	*n* (%)	*n* (%)	*n* (%)
Somatic concern	SOM	1.8 (1.2)	840 (58.4)	258 (17.9)	199 (13.8)	87 (6.1)	37 (2.6)	13 (0.9)	4 (0.3)
Anxiety	ANX	2.1 (1.2)	596 (41.4)	348 (24.2)	290 (20.2)	143 (9.9)	44 (3.1)	16 (1.1)	1 (0.1)
Emotional withdrawal	EMO	2.5 (1.5)	511 (35.5)	274 (19.1)	294 (20.4)	221 (15.4)	85 (5.9)	42 (2.9)	11 (0.8)
Conceptual disorganization	CON	2.4 (1.5)	599 (41.7)	251 (17.5)	265 (18.4)	176 (12.2)	102 (7.1)	32 (2.2)	13 (0.9)
Guilt feelings	GUI	1.5 (0.9)	1016 (70.7)	237 (16.5)	140 (9.7)	32 (2.2)	7 (0.5)	4 (0.3)	2 (0.1)
Tension	TEN	2.0 (1.1)	656 (45.6)	336 (23.4)	286 (19.9)	121 (8.4)	31 (2.2)	8 (0.6)	0 (0.0)
Mannerism and posturing	MAN	1.5 (1.0)	1090 (75.8)	148 (10.3)	117 (8.1)	56 (3.9)	18 (1.3)	9 (0.6)	0 (0.0)
Grandiosity	GRA	1.5 (1.0)	1107 (77.0)	140 (9.7)	97 (6.7)	63 (4.4)	14 (1.0)	12 (0.8)	5 (0.3)
Depressive mood	DEP	1.8 (1.1)	804 (55.9)	301 (20.9)	233 (16.2)	74 (5.1)	14 (1.0)	9 (0.6)	3 (0.2)
Hostility	HOS	1.9 (1.3)	807 (56.1)	256 (17.8)	169 (11.8)	136 (9.5)	47 (3.3)	15 (1.0)	8 (0.6)
Suspiciousness	SUS	2.4 (1.5)	587 (40.8)	248 (17.2)	269 (18.7)	201 (14.0)	81 (5.6)	41 (2.9)	11 (0.8)
Hallucinatory behavior	HAL	2.6 (1.6)	530 (36.9)	234 (16.3)	259 (18.0)	196 (13.6)	118 (8.2)	80 (5.6)	21 (1.5)
Motor retardation	MOT	1.7 (1.2)	898 (62.4)	244 (17.0)	158 (11.0)	82 (5.7)	37 (2.6)	15 (1.0)	4 (0.3)
Uncooperativeness	UNC	1.9 (1.2)	814 (56.6)	273 (19.0)	182 (12.7)	103 (7.2)	38 (2.6)	22 (1.5)	6 (0.4)
Unusual thought content	THO	2.5 (1.6)	578 (40.2)	239 (17.5)	251 (17.5)	181 (12.6)	118 (8.2)	54 (3.8)	17 (1.2)
Blunted affect	BLU	2.3 (1.4)	572 (39.8)	285 (19.8)	285 (19.8)	162 (11.3)	84 (5.8)	43 (3.0)	7 (0.5)
Excitement	EXC	1.6 (1.2)	1001 (69.6)	184 (12.8)	113 (7.9)	77 (5.4)	49 (3.4)	13 (0.9)	1 (0.1)
Disorientation	DIS	1.4 (0.8)	1144 (79.6)	162 (11.3)	87 (6.1)	30 (2.1)	7 (0.5)	3 (0.2)	5 (0.3)

BPRS, Brief Psychiatric Rating Scale; SD, standard deviation.

## Data Availability

Data sharing not applicable.

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
