# Peer review of "Network Analysis-Based Disentanglement of the Symptom Heterogeneity in Asian Patients with Schizophrenia: Findings from the Research on Asian Psychotropic Prescription Patterns for Antipsychotics"

_jpm, 2022, doi:10.3390/jpm12010033_

Round 1
Reviewer 1 Report
In this paper, the authors constructed a network structure of the Brief Psychiatric Rating Scale (BPRS) items in 1,438 Asian patients with schizophrenia. They found that motor retardation was situated most centrally within the BPRS network structure, followed by depressive mood and unusual thought content. Contrastingly, hallucinatory behavior was situated least centrally within the network structure. Using a community detection algorithm, the BPRS items were organized into positive, negative, and general symptom clusters. Overall, DSM symptoms were not more central than non-DSM symptoms within the symptom network of Asian patients with schizophrenia. Thus, motor retardation, which results from the unmet needs associated with current antipsychotic medications for schizophrenia, may be a tailored treatment target for Asian patients with schizophrenia. These findings are both interesting and important and will contribute to the field.
Overall, this paper is written in professional English with sufficient introduction, detailed methods and solid data. However, the discussion is relatively weak and needs to be further strengthened. The authors briefly discussed the neural mechanisms underlying the psychomotor retardation, including substantia nigra-based subcortical-cortical motor circuit, default-mode and sensory networks, and global cortical activity. But in fact, in addition to these networks, other important neural pathways can also be involved in the cognitive, sensory, and motor deficits in the patients -- the pulvinar nucleus of the thalamus in particular. Pulvinar is the largest nucleus in the thalamus and mutually connected with many cortical and subcortical regions, including the prefrontal cortex, sensory cortex, superior colliculus and amygdala (PubMed ID (PMID): 29175956). Thus, it plays important roles in normal multisensory processing, emotional response and decision making (PMID: 32142411; PMID: 31812514; PMID:26898778), which is significantly impaired in the schizophrenia patients. Also, the dysfunction of pulvinar is associated with cognitive and sensorimotor deficits seen in the the schizophrenia patients (PMID: 34583308; PMID: 33437351). The authors should include these important findings in the discussion.
I would like to recommend it to the editors after the revision.
Author Response
We greatly appreciate your kind comments.
According to your comments, we have revised the manuscript as follows:
We have added the contents in the discussion.
In addition to these networks, the pulvinar nucleus of the thalamus is an important neural structure that can be involved in cognitive, sensory, and motor deficits in patients with schizophrenia. The pulvinar nucleus is the largest nucleus in the thalamus and is mutually connected to several cortical and subcortical regions, including the prefrontal cortex, sensory cortex, superior colliculus, and amygdala [52]. Thus, the pulvinar nucleus plays an important role in normal multisensory processing, emotional response, and decision making, which are significantly impaired in patients with schizophrenia [53-55]. A positive relationship has been reported between performance on working memory and activation in the pulvinar nucleus and other structures [56]. Herein, structural and functional abnormalities of the pulvinar nucleus may be a neurobiological underpinning for the motor retardation-centered symptom networks observed in our study.
Reviewer 2 Report
Joonho Choi et. al. manuscript entitled “Network analysis-based Disentanglement of the Symptom Heterogeneity in Asian Patients’ with Schizophrenia: Findings from the Research on Asian Psychotrophic Prescription Patterns for Antipsychotics” is significant study and provides the data on Asian population with schizophrenia. The study supports that schizophrenia patients have motor retardation associated with the antipsychotic drugs and thus using or developing treatment regimen which targets non-Dopaminergic pathway is important. Methods, Results and Discussions are well-written and I suggest to accept the manuscript in present form.
Author Response
We greatly appreciate your kind comments.